# Inter- and Intra-Examiner Reliability Study of Two-Point Discrimination Test (TPD) and Two-Point Estimation Task (TPE) in the Sacral Area of Pain-Free Individuals

**DOI:** 10.3390/diagnostics13223438

**Published:** 2023-11-13

**Authors:** Edward Saulicz, Oskar Saulicz, Jakub Koterba, Damian Sikora, Aleksandra Saulicz, Mariola Saulicz

**Affiliations:** 1Institute of Physiotherapy and Health Sciences, Jerzy Kukuczka Academy of Physical Education, 40-065 Katowice, Poland; 2NZOZ “MED8” Miechowice, 41-908 Bytom, Poland; 3Bytom Medical Centre “Jedynka”, 41-902 Bytom, Poland; 4School of Public Health and Social Work, Queensland University of Technology (QUT), Kelvin Grove, QLD 4059, Australia

**Keywords:** tactile acuity, two-point discrimination test, two-point estimation task, sacral area, inter-examiner reliability, intra-examiner reliability

## Abstract

Tactile acuity is typically measured by a two-point discrimination test (TPD) and a two-point estimation task (TPE). In the back area, they are only conducted in the lumbar and cervical regions of the spine. Considering that such measurements have not been conducted in the sacral regions, the purpose of this study was to assess the inter- and intra-examiner reliability of the TPD and TPE at the level of the S3 segment. The study included 30 pain-free subjects aged 20–30 years. Tests were performed with a pair of stainless hardened digital calipers. The TPD was measured in two locations: 5 and 15 cm from the midline; for TPE both, points were located inside the measured area. Session 1 involved assessments by two examiners in 10-min intervals. Session 2 was measured by one examiner, at analogous intervals between tests. The TPD inter-rater reliability was excellent for mean measurements (ICC_3.2_: 0.76–0.8; ICC_3.3_: 0.8–0.92); the intra-rater reliability was excellent for mean measurements (ICC_2.2_: 0.79–0.85; ICC_2.3_: 0.82–0.86). The TPE inter-rater reliability was good to excellent for mean measurements (ICC_3.2_: 0.65–0.92; ICC_3.3_: 0.73–0.94); the intra-rater reliability for all studies (ICC_2.1_, ICC_2.2_, ICC_2.3_) was excellent (0.85–0.89). Two measurements are sufficient to achieve good reliability (ICC ≥ 0.75), regardless of the assessed body side.

## 1. Introduction

Tactile acuity is used as a marker of pathological conditions such as: chronic neuropathy, chronic neck and back pain syndromes, pain associated with degenerative joint changes, Sudeck syndrome, and temporomandibular joint dysfunction [1,2,3,4,5,6,7,8]. Research results have shown that the acuity with which touch is perceived at the painful body site is disturbed [9]. Studies have shown that tactile acuity in the acute phase of back pain only changes minimally during a 6-month follow-up [10]. Also, experimentally induced acute back pain leads to a loss of acuity, expressed as deteriorated touch sensation in the painful part of the body [11]. Such deficiencies can be diagnosed and quantitatively measured by performing relatively simple tests that are used to assess tactile acuity. Tactile acuity is most frequently measured by means of three different tests: a two-point discrimination test (TPD), a two-point estimation task (TPE), and a point-to-point test (PTP) [1,12,13,14,15,16]. The TPD test is the most common for tactile acuity, and although it is difficult to perform, the PTP is slightly more reliable, but it is not possible to perform it on all parts of the body, whereas the TPE task is characterized by the highest reliability, but appears to be susceptible to the test learning effect [9,11,13,16]. Therefore, in order to reliably reflect the tactile acuity disturbances in the lumbar spine, it is valid to use at least two of the three tests.

It has been assumed that pain located within the pelvis with mechanical causes is divided into three main categories: pregnancy-related pelvic pain, specific pathology pain, and pain of other origin [17,18,19]. Their common denominator is a disturbance in the function of the sacroiliac joints. Pelvic pain is obviously a highly complex phenomenon, and its causes are seen in structures other than the sacroiliac joints located within the pelvis itself, and it can be projected from the neighboring parts of the body [19,20,21]. The pain located deep within the gluteal area is referred to as deep gluteal syndrome, which can be caused by the compression of either the sciatic or pudendal nerves due to non-discogenic pelvic lesions [21]. The symptoms of deep gluteal syndrome can be related to the occurrence of pain and dysfunctional syndromes such as: piriformis syndrome, gemelli-obturator internus syndrome, ischiofemoral impingement syndrome, and proximal hamstring syndrome [19]. Under clinical conditions, deep gluteal syndrome appears as pain that occurs during physical activity related to hip bending, such as sitting or walking. Pelvic dysfunctional syndromes can be accompanied by different degrees of dysesthesia, which assumes different degrees of intensity, from numbness to hypoesthesia and hyperesthesia [22,23,24]. The measurement of tactile acuity can, therefore, be of clinical significance not only for the assessment of the current condition of the patient, but also for the assessment of pain and dysfunctional symptom remission in the treatment process.

Thus far, no scientific reports assessing tactile acuity at the pelvic level could be identified. Therefore, the purpose of this study was to assess the reliability of the tactile precision tests performed symmetrically on both sides of the lower part of the torso at the level of the sacrum.

## 2. Materials and Methods

The study was carried out on the premises of the Jerzy Kukuczka Academy of Physical Education in Katowice, at the Kinesiotherapy Department. The local Bioethics Committee provided approval for the study (no. 8/2016). Neither the participants nor the examiners were not informed of the purpose of the study. Every individual participating in the study was obligated to sign an informed consent declaration to participate in the study according to the guidelines of the Helsinki Declaration developed by the World Medical Association. The research was based both on reproducibility (two days apart) and on a repeatability (10 min apart) design [25].

### 2.1. Participants

Potentially healthy people were recruited for the study via oral announcement. The inclusion criteria contained normal healthy male and female adults aged between 20 and 30 years. The exclusion criteria included current back pain or back pain in the last three months, neurological disease, diabetes, previous surgical procedures in the lower back and pelvic area, painful menstruation, pregnancy, childbirth within the last year, and age >30 or <18. Such criteria were met by 37 people who expressed their willingness to participate in the study. Of this group, 7 subjects were excluded from the analysis as they could not participate in the second examination conducted 7 days after the first examination due to a date conflict, menstruation pain, the occurrence of pain in the lower back, or issues with detected light touch. Ultimately, 12 women and 18 men participated, aged between 20 and 30 (21.6 ± 2.04), between 159 to 193 cm in height (173.6 ± 8.7 cm) and with a body weight between 45 to 102 kg (69.87 ± 15.2 kg). The study group of 30 individuals corresponded to the criteria by Catley et al. [26] regarding the minimum number of subjects for the purpose of intra-examiner (18 subjects) and inter-examiner (14 subjects) studies in order to obtain 80% power of reliability.

### 2.2. Examiners

The study was carried out by a physician with 4 years of professional experience (Examiner A—OS) and a physiotherapist with 2 years of professional experience (Examiner B—JK). Both examiners were previously trained in TPD and TPE tests, and as of the commencement of the study they had experience in conducting such tests on healthy people and in clinical conditions (each examiner had performed TPD and TPE on at least 200 people). The sequence of the tests performed (TPD or TPE), their site (5 or 15 cm for TPD), and the side of the body (right, left) on which the examination was commenced was drawn by a person not directly involved in performing the measurements (MS). The same person read the measurement results from the electronic displays and saved them to a result sheet specifically prepared for this purpose.

### 2.3. Measurement Tools

The basic tool utilized for the TPD and TPE testing was a pair of stainless hardened digital calipers with a measurement range from 0 to 150 mm. In the case of the TPD test, one pair of calipers was required, whereas for TPE, two pieces were used. Both models featured a digital display that measured the caliper arm span with an accuracy of 0.01 cm. The other tool used to conduct the study was a therapeutic bed by Meden-in-med.

### 2.4. Position and Measurement Site

After conducting a medical history related to the assessment of possible contraindications to the test, collecting basic biometric data, and following the randomization of the order of tests and the side from which the test was started, the participants went to a separate room with a therapeutic bed on which they were instructed to lie front down, with an exposed back. A therapeutic roller was placed on the dorsal side of the feet to relax the muscles of the back side of the lower limbs. Subsequently, the examiner ordered the subjects to place their face in the opening of the head portion of the therapeutic bed. This was significant because, throughout the test, the examiner had to maintain free oral communication with the tested person. After the examiner made sure that the examined individual did not perceive any discomfort and that no discomfort sensations were present relating to maintaining this position for a prolonged period, the determination of the measurement sites could be commenced.

At the very beginning, the examiner found the third sacral segment (S3) by palpation on the medial sacral crest and marked it with a marker. The orientation point for finding the S3 segment was the upper ridge of the intergluteal cleft. Subsequently, by means of the calipers, the examiner measured 5 cm to the right and 5 cm to the left of the aforementioned point, marking both points. The last measurement was the measurement of 15 cm from the S3 point, both to the right and left using the calipers and marking these two points (see Figure 1).

### 2.5. Randomisation

Randomization concerned the spot where the study was started, previously specified in the measurements (left part of the sacrum at a distance of 5 cm, left part of the sacrum at a distance of 15 cm, right part of the sacrum at a distance of 5 cm, right part of the sacrum at a distance of 15 cm), and the order in which the TPD or TPE test was performed. For example, an individual for whom TPD was drawn by lot as the first test, on the right side and at a distance of 5 cm, was first subjected to the TPD test starting from the right side at a distance of 5 cm, followed by 15 cm. Subsequently, the examiner conducted the same test in the same order on the left side. After performing the TPD test, the examiner commenced the TPE test, starting with the right side and finishing on the left side. To maintain the randomness of the test parameters on the next subject, the measurements were started with the TPE test performed on the side of the body that was randomly selected, and then the TPD test was performed, starting from the same side from which the TPE test was started, with the difference that the distance was selected at random from the midline for the first measurement (5 or 15 cm). The person performing the drawing, after introducing the subject to the room where the test was performed, handed the examiner a piece of paper with the order of tests to be performed, the sides of the body, and measurement sites for the TPD test. This person made sure that, in subsequent tests (in the measurement performed by the second examiner after 10 min and in tests performed after 7 days at 10-min intervals by the same examiner), the same order of tests was followed (see Figure 2). The method of selecting the order of tests performed and the selection of time intervals between measurements (10 min, 1 week) were based on the research protocol used in our previous study [12].

In session 1 of the study, Examiner A (OS) left the room where the measurements were taken after the examination, and the study assistant (MS) invited Researcher B (JK), who was waiting in a separate room, to the testing room. Researcher B, after reading the note on the type of test (TPD or TPE), sites, and sites of measurement, then repeated the entire test procedure. In the follow-up test (session 2 was carried out 7 days after session 1), Examiner A performed measurements in the same order of test performance, body side, and location (for TPD).

### 2.6. TPD Test Performance

This test was carried out with a pair of calipers (stainless hardened, digital calipers, 150 mm) as discussed in the “Measurement tools” chapter. Before the main part of the test was carried out, the tested individual was instructed by the examiner about what the test consisted of and what wording should be used in the presented situation: “*I will touch your back area with a caliper. Your task will be to assess whether you feel the touch in one or two points. Then, you will provide one of the following answers: ONE or TWO. If, at the given moment, you are unsure of how many points you felt on your back, then please provide the answer I DON’T KNOW, and we will repeat the measurement in exactly the same spot*”. However, the person performing the test did not provide information on the purpose of this test in order to not suggest the potential behavior of the tested people, which could have affected their results. Subsequently, the examiner proceeded to begin the main part of the TPD test. The examiner applied the caliper arms to the back of the examined person at the previously determined point of S3, e.g., on the right side, with the force producing a slight bending of the skin in this area. Each touch with the calipers was to be linked with a response from the subject whether the touch was a single (ONE), a double (TWO), or if it was not appropriately interpreted (I DON’T KNOW). Following each touch and response provided, the examiner increased the distance of the caliper arms by 5 mm. The examiner increased this distance until the subject provided the response TWO, meaning that the touch was perceived in two points. Subsequently, the examiner applied the caliper arms again in the same spot, and if he received the response TWO, then one arm of the calipers was applied to the back of the subject. If the subject provided the answer ONE, then the examiner recorded the obtained result in the previously prepared tables. Subsequently, the examiner began the second version of the TPD test, in which the initial position was the spacing of the caliper arms to the distance obtained in the first measurement, increased by 10 mm. This time, the examiner reduced the distance between the caliper arms by 5 mm after each touch, until the subject provided the response ONE. When the subject provided the response ONE, as in the preceding TPD test, the examiner once again applied the caliper arms in the spot where one point was sensed by the subject. A repeated ONE answer resulted in recording the obtained distance in the result table. The averaged value from these two measurement methods constituted the result of the first trial. The entire cycle was repeated three times (interchangeable with three measurements with increasing the distance between the caliper arms and three measurements with its decreasing). In the same manner, measurements were taken at a distance of 5 cm, as well as 15 cm from the S3 point marked on the sacrum. The same procedure was performed on both sides of the body. After the measurements were obtained and the results were recorded in the table, the examiner disinfected the calipers.

### 2.7. TPE Test Performance

Two pairs of calipers (stainless hardened, digital calipers, 150 mm) were used for the TPE test. One of the pairs of calipers was held by the examiner, the other was provided to the subject. The subject was holding the calipers in such a manner that the digital display was turned away from the subject. Similar to the TPD test, the examiner also explained the procedure of the test and instructed what expressions he expected from the subject: “*I will touch your back area with the caliper arms in such a manner that you will feel the touch on the back in two different points. Your task will be to use your caliper to estimate the distance between the perceived points as accurately as you can in such a manner that you spread the caliper arms to the distance you believe to correspond to the sensation on your back. Throughout the test, you must not look at the display showing the estimated distance. After each measurement, present the value on the display to the assistant recording the results*”. Subsequently, the examiner set the caliper at a distance of 120 mm, which according to previous tests, perfectly corresponds with the sensation of two separate points [10,15]. The next step was to put the caliper on the tested person at the S3 level on one of the sides (drawn earlier), in such a way that the previously marked points (5 and 15 cm) were between the spaced caliper arms. When the examined person most accurately reproduced the result by setting the caliper arms, they showed the result recorded on the digital display to the person assisting the examiner, who entered the result into a previously prepared table. After each measurement, the subject reset the caliper arms so that they were connected. The same test procedure was carried out on the other side of the body. The measurements on each side were repeated three times. After the TPE test was carried out and the results were recorded in the table by the assistant, the examiner disinfected both pairs of calipers again.

### 2.8. Statistical Analysis

For the calculation of the intra- and inter-rater reliability, intraclass correlation coefficient (ICC) type 3,1, 3,2, 3,3 and type 2,1, 2,2, 2,3 were used, respectively. The ICC value was interpreted in the following manner: below 0.4, poor reliability; between 0.40 and 0.59, fair reliability; between 0.60 and 0.74, good reliability; and above 0.75, excellent reliability [27]. In addition, the standard error of measurement (SEM = SD × √1 − ICC). Data were analyzed using STATISTICA 13.3 PL (StatSoft Inc., Tulsa, OK, USA) and Windows Microsoft Excel version 2007 (Microsoft Corporation, Redmond, WA, USA) software.

## 3. Results

The averaged values in millimeters of the subsequent TPD and TPE measurements are included in Table 1. For both examiners (in session 1) and both tests performed by one examiner (session 2), independently of the examined side of the body, higher perception precision was recorded at the measurement site located 5 cm from S3.

### 3.1. Reliability of Two-Point Discrimination (TPD) Test—One Examiner

The analysis of the data in Table 2 shows that the two-point discrimination test performed by one examiner at 10 min after the baseline measurement in the closer site (TPD 5 centimeters) showed good reliability on both sides (ICC = 0.67–74; 95% CI: 0.42–87) for single measurements (ICC_3.1_), and the corresponding SEM values were 0.003–0.5. Excellent reliability was obtained for the mean measurements (ICC_3.2_ = 0.76–0.8; 95% CI: 0.55–90 and SEM 0.01–0.33, and ICC_3.3_ = 0.8–0.82; 95% CI: 0.63–91 and SEM 0.04–0.27). Analogous measurements carried out in the distant location (TPD 15 centimeters) showed fair to excellent reliability (ICC = 0.59–0.78; 95% CI: 0.30–89) for single measurements (ICC_3.1_), with SEM values of 0.41. Excellent reliability was obtained for the mean measurements (ICC_3.2_ = 0.76–0.8; 95% CI: 0.51–93 and SEM = 0.01–0.33, and ICC_3.3_ = 0.8–0.92; 95% CI: 0.56–96 and with SEM values 0.22–0.27).

In the study carried out by the intra-examiner one week from the first measurement (Table 2) for the TPD test at the close site (5 cm), it showed good reliability (ICC = 0.61–0.63; 95% CI: 0.33–81 and SEM = 0.17–0.21) for single measurements (ICC_3.1_). For the means of two and three measurements (ICC_3.2_ and ICC_3.3_), good to excellent reliability was obtained (ICC = 0.71–0.75; 95% CI: 0.47–87, with SEM values 0.06–0.32). The TPD measurement carried out at a distance site (15 cm) for just single measurements (ICC_3.1_) showed good to excellent reliability (ICC = 0.67–0.76), and the corresponding SEM values were 0.28–0.33). Good to excellent reliability was obtained for the mean measurements (ICC_3.2_ = 0.69–0.78; 95% CI: 0.44–0.89 and SEM = 0.02–0.005, and ICC_3.3_ = 0.68–0.77; 95% CI: 0.42–89 with SEM values 0.04–0.06).

### 3.2. Reliability of Two-Point Discrimination (TPD) Test—Two Examiners

The analysis of the data in Table 3 shows that the two-point discrimination test performed by two examiners 10 min after the baseline measurement at the closer site (TPD 5 centimeters) showed good to excellent reliability (ICC = 0.71–0.76; 95% CI: 0.32–0.85 and the corresponding SEM values were 0.04–0.26) for single measurements (ICC_2.1_). Excellent reliability was obtained for the mean measurements (ICC_2.2_ = 0.79–0.85; 95% CI: 0.60–0.92 and SEM = 0.13–0.34, and ICC_2.3_ = 0.82–0.86; 95% CI: 0.63–0.93), ICC_2.3_ (0.82–0.86 with SEM values 0.17–0.38). For the measurements performed 15 cm from the longitudinal axis of the sacrum (TPD 15 cm), the reliability of the measurements between the two investigators for all measurements (ICC_2.1_, ICC_2.2_, ICC_2.3_) was excellent (ICC = 0.77–0.85; 95% CI: 0.57–0.93 and the corresponding SEM values were 0.02–0.27).

### 3.3. Reliability of Two-Point Estimation (TPE) Test—One Examiner

Reliability of single measurements (ICC_3.1_). The two-point estimation (TPE) test was excellent for direct (10 min) measurement (ICC = 0.83–0.88; 95% CI: 0.68–0.94 with SEM values 0.82–0.85) (Table 4). Excellent reliability was obtained for the mean measurements (ICC_3.2_ and ICC_3.3_) (ICC_3.2_ = 0.88–0.92; 95% CI: 0.77–0.96 with SEM values 0.58–0.69, and ICC_3.3_ = 0.88–0.94; 95% CI: 0.77–0.97 and SEM = 0.49–0.75). For the late TPE measurement (performed 7 days after the first examination), the reliability of the single measurements (ICC_3.1_) was fair to good (ICC = 0.59–0.69; 95% CI: 0.29–0.84 with SEM values 0.95–1.04). The means of the two and three measurements (ICC_3.2_ and ICC_3.3_) showed good to excellent reliability (ICC_3.2_ = 0.65–0.72; 95% CI: 0.49–0.86 and the corresponding SEM values were 0.22–0.51, and ICC_3.3_ = 0.73–0.75; 95% CI: 0.51–0.87 with SEM values 0.17–0.51).

### 3.4. Reliability of Two-Point Estimation (TPE) Test—Two Examiners

The reliability of the measurements between the two investigators for all studies (ICC_2.1_, ICC_2.2_, ICC_2.3_) was excellent (ICC values were 0.85–0.89; 95% CI: 0.71–0.95 and the corresponding SEM values were 0.78–1.41) (Table 5).

## 4. Discussion

Thus far, no assessment of the intra- and inter-examiner reliability of the TPD and TPE has been conducted on the sacral spine. Therefore, it is not possible to refer the results of this study to the results of other researchers. Therefore, the purpose of this study was to assess the reliability of the tactile precision tests symmetrically performed on both sides of the lower part of the torso at the level of the sacrum.

The results concerning the assessment of the TPD accuracy in healthy individuals show that, independent of the measurement site (5 cm or 15 cm from the central axis of the sacrum, to its right or left), the intra- and inter-rater reliability were good to excellent. Similar reliability (good to excellent) was recorded for TPE, and in the case of the measurements taken by two examiners, the inter-rater reliability was excellent. In the case of both tests, two repetitions of the measurement was sufficient to obtain high reliability of the results (ICC ≥ 0.75).

Assessment of the TPD and TPE measurement reliability has thus far been performed on the lumbar spine. Only in two studies were the measurements taken at the L5 level, and thus in the area closest to the measurement level applied in the protocol of our study. In a study by Ehrenbrusthoff et al. [13], excellent reliability (ICC = 0.8) was recorded for one examiner and fair reliability (ICC = 0.53) for two examiners. However, this assessment only concerned people with non-specific CLBP. Wang et al. [28] investigated the reliability of the TPD on the painful and non-painful sides in patients with CLBP. In contrast to our study, these authors separately assessed the TPD with increasing and decreasing spacing of the caliper arms. The authors, for this test on the left healthy side, obtained a value of ICC = 0.74–0.82 in young people (aged 18–35) and ICC = 0.76–0.78 for older people (aged 36–65) for one examiner and ICC = 0.66–0.75 in young people and ICC = 0.72–0.78 in older people for two examiners. In our study, similar results were obtained for a single TPD measurement (ICC_3.1_ = 0.67–74) performed by one examiner and measurements performed by two examiners (ICC_2.1_ = 0.71–0.76) on the closer site (5 cm, and thus at an analogous distance from the spine axis as used by the mentioned authors for the measurements). Wang et al. [28] also assessed the reliability of TPE tests, which, on the healthy side in both age groups of the examined participants, was high for one examiner (ICC = 0.82–0.85) and slightly lower in the measurements carried out by two examiners (ICC = 0.70–73). In our study, similar reliability was obtained for a single TPE examination made by one examiner (ICC_3.1_ = 0.83–0.88) and was slightly superior (ICC_2.1_ = 0.85–0.86) for two examiners. The averaged values from consecutive measurements (second and third) in our research provided slightly more precise results of both analyzed tests. Without a doubt, the results of the present study were affected by the fact that the subjects were only young and healthy people.

Assessment of the accuracy of tactile acuity measurements in the sacral area of a pain-free subject should be treated as a type of pilot study expanding the field of exploration of the superficial perception assessment with the pelvic region. On this basis, it can only be stated that the inter- and intra-examiner reliability in healthy people is similar to the measurement reliability in the neighboring lumbar spine [9]. At this stage, it is difficult to judge to what extent the measurement of the precision of tactile sensation performed at the cross level will prove successful in clinical conditions. In the lumbar spine region, most of the horizontal TPD and TPE measurements are made at the same dermatome level. In our study, the measurement performed in the proximal place, where it was started at a distance of 5 cm from the midline of the S3 segment, included the S2 and S1 dermatomes and the measurement of TPD, and which started at a distance of 15 cm, could include up to three dermatomes (S2-S1-L5). Is this in fact related to the less precise touch sensation recorded in the TPD test circumferentially performed at a distance of 15 cm from the center of the S3 segment (10.8–18.4 mm on the right side and 8.8–15.2 mm on the left side—Table 1); or, is the tactile sensation at the pelvic level somewhat neurogeometric in nature and characterized by greater precision in the midline of the body? Research on body orientation in space and vibrotactile localization has indicated that the midline of the body is more important than other parts of the body [29,30,31]. Naturally, such speculations regarding tactile acuity require research-based confirmation or falsification, and a reliable tool to base such research on is the measurement of TPD and TPE, as indicated by the results of our research.

Patients with lower back pain (LBP) are characterized by poorer tactile acuity in the painful region as compared to healthy individuals [9,14,16,32]. In the case of chronic LBP, the central nervous system differently reacts to tactile stimuli because greater activity is observed in the sensory cortex in the areas responsible for the perception of pain and reduced activity in the structures responsible for inhibiting pain pathways [32,33,34,35,36,37]. The research showed that such changes result in the perception of the body. Patients with chronic LBP have a disturbed pattern of the painful part of the body in which they describe their painful back in two ways, either as diminished or enlarged compared to the non-painful parts of the body [38,39,40,41]. The role of tactile disturbances and body perception in strengthening pathological behavior patterns remains unknown. Changes in motor planning most likely occur immediately after acute pain in the lower back, and pelvic girdle occurs and can therefore be a natural response to a sense of distress. Disturbed sensation and inadequate perception of the painful part of the back may play an important role in the development of pathological patterns of motor behavior, in essence activating a type of circum vitiosus mechanism of mutual feedback favoring the chronicization of ailments. The pelvis can be assigned significance in this case, as it determines the spatial layout of the torso and thus it is referred to as the key to body posture. In clinical practice, the so-called lateralization of the torso with a lateral shift of the pelvis is often observed for a long time after the relief of acute pain symptoms.

The abovementioned cortical reorganization, which changes the perception of the painful area of the body, can be a significant obstacle to recovery. The plasticity of the brain underlying the reorganization of the cerebral cortex also indicates that it may be a reversible process, but is dependent on targeted treatment [42]. The first attempts at such activities turned out to be so encouraging that, over time, it led to the development of sensory discrimination training [43,44,45,46,47]. In this training, the patient recognizes the location and type of stimuli used based on feedback. Studies have shown that such training improves tactile perception, which results in a reduction in back and limb pain [44,48]. Interesting observations were provided in the research by Kim et al. [32]. During a parallel assessment of the structure of the primary somatosensory cortex S1 and tactile acuity, it allowed the authors to observe not only the relationship between the deterioration of the tactile acuity in people with chronic LBP, with changes in the gray and white matter in the primary somatosensory cortex, but also that a 4-week acupuncture therapy caused early mechanical changes in somatosensory processing, which resulted in improved touch acuity [32]. This fact opens new possibilities for using TPD and TPE tests not only to assess the current clinical condition, but also to assess the effectiveness of various methods of LBP therapy.

The results of the TPD and TPE tests performed at the 7-day interval present moderate/moderately high reliability. It is sufficient for conducting research in a longer time frame, although it is characterized by a slightly lower repeatability compared to the tests directly carried out (after 10 min). This distribution of results could have been influenced by environmental factors (e.g., ambient temperature, time of day, changes in atmospheric pressure), changes in the patient’s mental status (emotions, agitation, fatigue, etc.), and other factors that may have an impact on the change in tactile acuity. In summary, it can be concluded on the basis of the obtained research results that the TPD and TPE tests are reliable methods for measuring tactile acuity in the sacral back section (at the level of the S3 segment) in healthy people. The reliability of two measurements is sufficient to measure tactile acuity in a healthy person, regardless of whether the measurement is performed by one or two examiners, and the longer time interval (7 days) is not a factor that significantly deteriorates the precision of the measurement.

Considering that the study only included healthy women and healthy men, a certain limitation of the study was the lack of taking into account the phase of the cycle among the studied group of women. As is known, in the second phase of the cycle, water is retained in the female body, including subcutaneous tissue, which changes the conditions for the sensitivity of the sense of touch [49,50]. In the second phase of the cycle, tactile hyperesthesia temporarily occurs. Future research should be planned so that the later measurement of the tactile testing in the sacral area in women occurs in the same cycle phase.

## 5. Conclusions

Both the TPD and TPE tests are reliable methods for measuring tactile acuity in the sacral spine (at the S3 level) in healthy people. In measurements conducted in a short time interval (10 min) for results obtained by one researcher (intra-examiner reliability), as well as by two researchers (inter-examiner reliability), to obtain high reliability of results (ICC ≥ 0.75), two measurements are sufficient, regardless of the assessed side of the body, and place of measurement (5 or 15 cm). However, for the examinations carried out weekly, both tests exhibit moderate reliability. The TPD test performed after a 7-day break is characterized by average reliability, and increasing the number of repeated measurements does not significantly improve the reliability of this test. The TPE test repeated after a 7-day break requires three replications to achieve moderate or high reliability.

## Figures and Tables

**Figure 1 diagnostics-13-03438-f001:**
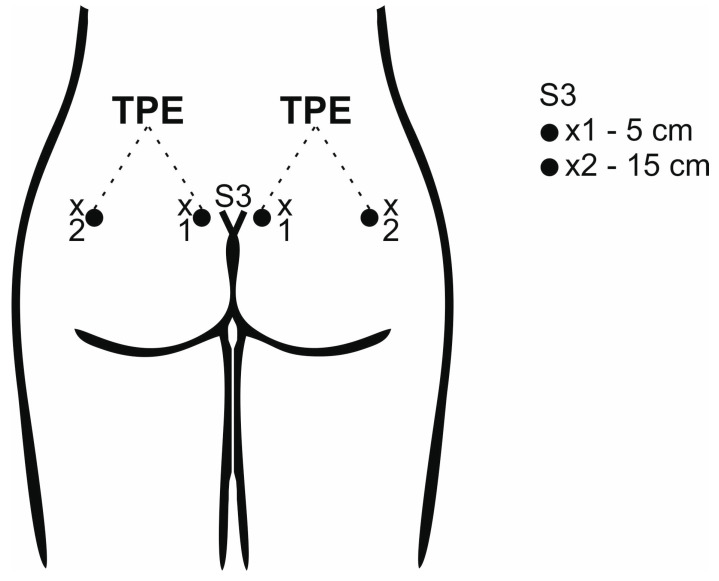
Example of the determination of the measurement points for the TPD (×1 and ×2) and TPE tests at the level of the third sacral segment.

**Figure 2 diagnostics-13-03438-f002:**
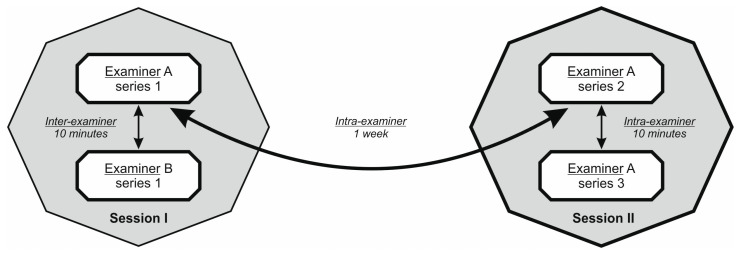
Structure of the study. The first session enrolled inter- and the second intra-examiner reliability.

**Table 1 diagnostics-13-03438-t001:** Measurement results (mean value; SD, IC 95%) TPD and TPE obtained by one examiner and by two examiners.

Examiner	Measurement	TPD [5 cm]		TPD [15 cm]		TPE	
		Right (mm)	Left (mm)	Right (mm)	Left (mm)	Right (mm)	Left (mm)
	1	45.1 (9.2) 41.7–48.5	46.8 (9.2) 43.3–49.9	60.1 (13.4) 55.1–65.1	58.6 (14.1) 53.4–63.9	71.3 (25.2) 61.9–80.7	72.8 (25.8) 63.2–82.5
A [session 1]	2	44.9 (8.8) 41.6–48.2	46.1 (10.2) 42.3–49.9	59.2 (12.1) 54.6–63.7	58.5 (12.1) 53.9–63.0	73.8 (23.3) 65.1–82.4	74.9 (22.1) 66.6–83.1
	3	45.5 (9.1) 42.1–48.9	46.1 (9.7) 42.5–49.8	59.4 (13.6) 54.4–64.5	58.8 (11.7) 54.4–63.2	76.2 (25.2) 66.8–85.6	74.7 (24.5) 65.6–83.9
	1	45.1 (10.3) 41.2–48.9	50.6 (9.7) 46.9–54.2	63.5 (15.4) 57.8–69.3	64.5 (13.4) 59.5–69.5	71.3 (24.1) 62.3–80.3	73.3 (24.7) 64.0–82.5
B [session 1]	2	47.7 (10.9) 43.6–51.8	49.4 (10.3) 45.6–53.3	63.2 (14.6) 57.8–68.7	64.6 (11.7) 60.2–68.7	73.8 (26.3) 64.0–83.7	73.6 (22.9) 65.0–82.1
	3	46.6 (11.1) 42.4–50.7	50.6 (9.7) 47.0–54.3	63.1 (13.9) 57.9–68.2	62.4 (13.4) 57.4–67.5	75.3 (25.5) 65.8–84.8	73.6 (23.7) 64.8–82.5
	1	46.7 (9.1) 43.3–50.1	48.9 (8.3) 45.8–51.9	60.4 (10.6) 56.4–64.3	59.9 (12.0) 55.4–64.4	78.8 (25.1) 69.4–88.2	80.5 (27.0) 70.4–90.6
A [session 2]	2	47.2 (8.9) 43.9–50.5	48.4 (9.1) 44.9–51.8	61.0 (10.6) 57.1–64.9	59.4 (12.5) 54.7–64.0	78.3 (25.1) 68.9–87.7	79.8 (26.7) 69.8–89.8
	3	47.4 (8.3) 44.3–50.5	48.9 (8.5) 45.8–52.1	60.2 (10.0) 56.4–63.9	59.7 (12.0) 55.2–64.2	79.0 (25.0) 69.7–88.3	78.4 (25.1) 69.0–87.8
	1	48.6 (7.4) 45.8–51.3	50.6 (8.3) 47.5–53.7	62.7 (9.2) 59.2–66.1	59.4 (9.5) 55.9–63.0	79.7 (27.2) 69.5–89.8	81.2 (29.5) 70.2–92.2
A [session 2]	2	49.1 (7.9) 46.1–52.0	50.2 (7.5) 47.4–53.0	59.9 (9.3) 56.4–63.4	61.7 (10.3) 57.8–65.5	81.9 (27.5) 71.6–92.1	81.6 (28.1) 71.1–92.0
	3	48.8 (8.0) 45.8–51.8	49.8 (7.7) 46.9–52.7	60.0 (9.1) 56.7–63.4	61.3 (10.1) 57.6–65.1	81.6 (28.1) 71.1–92.1	81.9 (27.8) 71.5–92.3

**Table 2 diagnostics-13-03438-t002:** Two-point discrimination (TPD) test and retest performed by intra-examiner (one examiner).

Side	Measurement	Distance	ICC_3.1_ (95% CI)	SEM	ICC_3.2_ (95% CI)	SEM	ICC_3.3_ (95% CI)	SEM
	direct	5 cm	0.67 (0.42–0.83)	0.5	0.76 (0.55–0.88)	0.33	0.8 (0.63–0.90)	0.27
		15 cm	0.59 (0.30–0.78)	0.41	0.74 (0.51–0.86)	0.35	0.76 (0.56–0.88)	0.35
Right	late	5 cm	0.61 (0.33–0.80)	0.21	0.75 (0.54–0.87)	0.32	0.75 (0.54–0.87)	0.25
		15 cm	0.67 (0.42–0.83)	0.33	0.69 (0.44–0.84)	0.02	0.68 (0.42–0.83)	0.004
	direct	5 cm	0.74 (0.52–0.87)	0.003	0.8 (0.62–0.90)	0.01	0.82 (0.65–0.91)	0.04
Left		15 cm	0.78 (0.60–0.89)	0.41	0.86 (0.74–0.93)	0.33	0.92 (0.83–0.96)	0.22
	late	5 cm	0.63 (0.36–0.81)	0.17	0.71 (0.47–0.85)	0.06	0.71 (0.48–0.85)	0.11
		15 cm	0.76 (0.56–0.88)	0.28	0.78 (0.58–0.89)	0.005	0.77 (0.58–0.89)	0.06

95% CI—95% Confidence Interval; direct—measurement taken after 10 min; late—measurement taken after 1 week.

**Table 3 diagnostics-13-03438-t003:** Two-point discrimination (TPD) test and retest performed by inter-examiner (two examiners).

Side	Distance	ICC_2.1_ (95% CI)	SEM	ICC_2.2_ (95% CI)	SEM	ICC_2.3_ (95% CI)	SEM
	5 cm	0.71 (0.47–0.85)	0.26	0.85 (0.71–0.92)	0.34	0.86 (0.73–0.93)	0.38
Right	15 cm	0.79 (0.61–0.89)	0.27	0.84 (0.70–0.92)	0.23	0.85 (0.70–0.93)	0.16
	5 cm	0.6 (0.32–0.79)	0.19	0.79 (0.60–0.89)	0.13	0.82 (0.63–0.91)	0.17
Left	15 cm	0.77 (0.57–0.88)	0.04	0.78 (0.58–0.89)	0.02	0.81 (0.63–0.91)	0.08

95% CI—95% Confidence Interval.

**Table 4 diagnostics-13-03438-t004:** Two-point estimation (TPE) test and retest performed by intra-examiner (one examiner).

Side	Measurement	ICC_3.1_ (95% CI)	SEM	ICC_3.2_ (95% CI)	SEM	ICC_3.3_ (95% CI)	SEM
	direct	0.83 (0.68–0.92)	0.82	0.88 (0.77–0.94)	0.69	0.88 (0.77–0.94)	0.75
Right	late	0.69 (0.44–0.84)	0.95	0.72 (0.49–0.86)	0.51	0.75 (0.54–0.87)	0.51
	direct	0.88 (0.77–0.94)	0.85	0.92 (0.84–0.96)	0.58	0.94 (0.87–0.97)	0.49
Left	late	0.59 (0.29–0.78)	1.04	0.65 (0.39–0.82)	0.22	0.73 (0.51–0.86)	0.17

95% CI—95% Confidence Interval; direct—measurement taken after 10 min; late—measurement taken after 1 week.

**Table 5 diagnostics-13-03438-t005:** Two-point estimation (TPE) test and retest performed by inter-examiner (two examiners).

Side	ICC_2.1_ (95% CI)	SEM	ICC_2.2_ (95% CI)	SEM	ICC_2.3_ (95% CI)	SEM
Right	0.86 (0.73–0.93)	0.78	0.85 (0.73–0.93)	0.83	0.87 (0.74–0.94)	0.8
Left	0.85 (0.71–0.93)	1.41	0.86 (0.73–0.93)	1.14	0.89 (0.79–0.95)	0.84

95% CI—95% Confidence Interval.

## Data Availability

The data that supports the findings of this study is available from the corresponding author, upon reasonable request.

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
