# Peer review of "Inter- and Intra-Examiner Reliability Study of Two-Point Discrimination Test (TPD) and Two-Point Estimation Task (TPE) in the Sacral Area of Pain-Free Individuals"

_diagnostics, 2023, doi:10.3390/diagnostics13223438_

Round 1

Reviewer 1 Report

Comments and Suggestions for Authors

The purpose of this study was to assess the reliability of the tactile precision tests performed symmetrically on both sides of the lower part of the torso at the level  of the sacrum.

L73.  Please describe the standards of reliability studies you followed. 

L149, The 7 days interval is valid? Please cite the previous study.

L346. “Patients with Lower Back Pain (LBP) are characterised by poorer tactile acuity in the 346 sore region as compared to healthy individuals” Is the sentence based on the your results?

Conclusions: it is too long. Please summarize it. 

Reviewer 2 Report

Comments and Suggestions for Authors

This research appears to be warranted. The project was well-designed and carried out, for example, detail in the methods would easily allow for replication. Most of my comments related to word choice and additional clarity. See specific comments on the PDF.

Comments on the Quality of English Language

Overall, English grammar acceptable, but I made some suggestions on word choices.
